# Plastic pollution fosters more microbial growth in lakes than natural organic matter

Eleanor A. Sheridan [1,2✉], Jérémy A. Fonvielle [1], Samuel Cottingham[1], Yi Zhang[1], Thorsten Dittmar [3,4], David C. Aldridge [2] & Andrew J. Tanentzap [1✉]

Plastic debris widely pollutes freshwaters. Abiotic and biotic degradation of plastics releases carbon-based substrates that are available for heterotrophic growth, but little is known about how these novel organic compounds influence microbial metabolism. Here we found leachate from plastic shopping bags was chemically distinct and more bioavailable than natural organic matter from 29 Scandinavian lakes. Consequently, plastic leachate increased bacterial biomass acquisition by 2.29-times when added at an environmentally-relevant concentration to lake surface waters. These results were not solely attributable to the amount of dissolved organic carbon provided by the leachate. Bacterial growth was 1.72-times more efficient with plastic leachate because the added carbon was more accessible than natural organic matter. These effects varied with both the availability of alternate, especially labile, carbon sources and bacterial diversity. Together, our results suggest that plastic pollution may stimulate aquatic food webs and highlight where pollution mitigation strategies could be most effective.

[1] Ecosystems and Global Change Group, Department of Plant Sciences, University of Cambridge, Cambridge CB2 3EA, United Kingdom. [2] Department of Zoology, University of Cambridge, The David Attenborough Building, Cambridge CB2 3QZ, United Kingdom. [3] Institute for Chemistry and Biology of the Marine Environment (ICBM), University of Oldenburg, 26129 Oldenburg, Germany. [4] Helmholtz Institute for Functional Marine Biodiversity at the University of Oldenburg (HIFMB), 26129 Oldenburg, Germany. ✉email: eleanorasheridan@gmail.com; ajt65@cam.ac.uk

The response of microbes to widespread and growing plastic pollution in freshwaters has consequences for ecosystem metabolism and food web health[1–3]. In addition to providing a substrate for biofilm colonisation[4], plastics leach dissolved organic matter (DOM) during mechanical, photochemical, and biological degradation[5–7]. This plastic leachate can provide energy for bacterial growth[8,9], and be transferred upwards through food webs to support the growth of higher trophic levels[10]. However, plastic leachate can also impair bacterial growth because of toxic compounds added to synthetic polymers during manufacturing, for example to increase plastic flexibility and heat stability[11]. As many of these toxic additives are hydrophobic organic compounds that tightly sorb to synthetic polymers, they can also harm, and potentially biomagnify in, higher trophic levels that ingest bacterial decomposers[2]. Determining the conditions in which bacteria can best grow, and consequently deplete plastic leachate from the environment, can ultimately help prioritise efforts to mitigate and clean-up global plastic pollution.

Few data exist on the molecular composition and fate of plastic leachate in freshwaters, especially compared with natural DOM. Synthetic polymers are generally regarded as non-biodegradable[12], but plastics also contain many labile and potentially bioavailable additives—such as plasticizers, colourants, and antioxidants—that are used to give polymers their functional properties[13–15]. These additives can account for up to 70% of plastic debris on a per-mass basis[14,15]. The most common plastics, i.e. polyethylene and polypropylene[16,17], are also buoyant and so undergo the highest rates of photodegradation and leaching in the warm, irradiated conditions of surface waters[9]. Consequently, plastic leachate can accumulate at high concentrations in surface waters relative to natural DOM[8]. If this leachate contains more labile compounds than natural DOM, bacteria should be able to grow and cycle nutrients more efficiently[18,19]. Structural differences between molecules in plastic leachate and natural DOM could similarly enhance bacterial growth by providing more niches for decomposition[20]. Previous studies[8,9,11] have shown how the response of bacteria to plastic leachate can vary, but, to our knowledge, no study has tested whether the molecular composition of DOM may explain this variation. Recent advances in ultra-high-resolution mass spectrometry now provide an opportunity to address this question[21–23].

The responses of bacteria to plastic leachate should vary across waters for at least two reasons. First, the molecular composition of natural DOM varies among lakes and rivers[24,25], and so should influence the ability of bacteria to use plastic leachate. In most of the world's lakes, DOM is dominated by relatively recalcitrant compounds[26,27], limiting opportunities for decomposition[20,28]. Plastic leachate that is more labile may therefore be widely assimilated in lakes containing this recalcitrant carbon. By contrast, leachate may have little benefit to bacteria in waters with already highly labile DOM, or it may be used similarly to natural DOM that it resembles chemically, as bacteria will be preadapted to use these substrates[29]. Second, the functional composition of bacterial communities, and thus their ability to utilise natural DOM, varies across space because of different environmental conditions, dispersal histories, and stochastic processes[30–33]. The same pattern should also be seen for DOM derived from plastic leachate.

Here, our aim was to determine the effects of plastic leachate on bacteria in the northern lakes that dominate the world's freshwater area[34]. We hypothesised that the molecular composition of pre-existing lake DOM controls how bacteria respond to plastic leachate. To test our hypothesis, we incubated surface waters from 29 lakes of varying DOM composition either with or without leachate from low-density polyethylene (LDPE) plastic bags, the most common plastic in freshwaters[35]. We added either an environmentally representative amount of this leachate (0.1 mg C L$^{-1}$; Supplementary Methods 1), which was much less than used in previous studies[8,9,11], or an identical volume of a distilled water control to surface lake water. Using Fourier-transform ion cyclotron resonance mass spectrometry (FT-ICR-MS), we compared the molecular composition of DOM in our plastic leachate to that naturally occurring in our study lakes. We also measured bacteria protein production (BPP), which reflects bacterial biomass acquisition[36], and bacterial growth efficiency (BGE). BGE allows us to separate whether BPP increases with leachate simply because more carbon is available or because the added carbon is also more labile and thus more accessible to bacteria. In the former case, BGE would remain unchanged, as any increase in BPP would purely result from an increase in the absolute amount of carbon processed rather than any change in how it was processed. In the latter case, BGE would increase with BPP because the carbon would be processed more efficiently, such as if it was more bioavailable to resident bacterial communities. We further tested how the responses of BPP and BGE to plastic leachate varied with microbial community structure and which taxa were associated with these responses using 16S amplicon sequencing. Our work now advances previous studies by showing that the effects of plastic leachate on BGE strongly depend on the concentration and functional diversity (FD) of existing lake DOM, thereby explaining variation in the responses reported to date[8,9,11].

## Results

**Plastic leachate is more labile than natural organic matter.** DOM from plastic leachate was distinct from that in lakes in three main ways. First, it had less diversity in the potential functions (i.e. reactivity) of molecular formulas. We used a widespread functional diversity (FD) index to calculate the expected mass difference between molecules in the dataset. The FD of plastic leachate was 3.46, lower than any of the 22 lakes in which we measured FD. In these lakes, FD ranged from 6.12 to 6.96, indicating more variation in the potential size range of molecules available for microbial activity. These differences were mirrored by the total number of molecular formulas that we detected in our analytical window (150–2000 Da): 855 in the leachate versus between 3684 to 7116 in natural lake DOM. Second, despite being less diverse, plastic leachate had a much higher lability index. Of the molecular formulas detected in the plastic leachate, 18.6% had a high lability index[21] (i.e. H:C ratio ≥1.5), exceeding proportions found in any of our 22 study lakes, which ranged from 10.3 to 12.5%. Although the lakes did have a greater absolute number of compounds with a high lability index given their larger number of molecular formulas, highly labile compounds were relatively less abundant (5.4–10.6%) in lakes than within plastic leachate where they accounted for 82.2% of the normalised peak intensity. Compared to a freshwater standard widely used in mass spectrometry, the plastic leachate also had a greater H:C ratio, a lower O:C ratio, fewer molecular formulas, and a greater percentage of formulas with a high lability index (Fig. 1). Finally, 35% of molecular formulas in the plastic leachate were unique and absent from our 22 study lakes. This value likely underestimated the true difference. Of our study lakes, we previously surveyed 19 for pollution impacts and all were contaminated with microplastics and anthropogenic fibres[37]. Thus, it is likely we detected associated plastic-derived compounds in DOM of these lakes. Our approach also resolved molecular formulas and not structures, so identical formulas between plastic leachate and lake DOM could represent different molecules. Irrespective, 11 of the formulas unique to the leachate corresponded to known chemical additives used in plastic

## Plastic Leachate

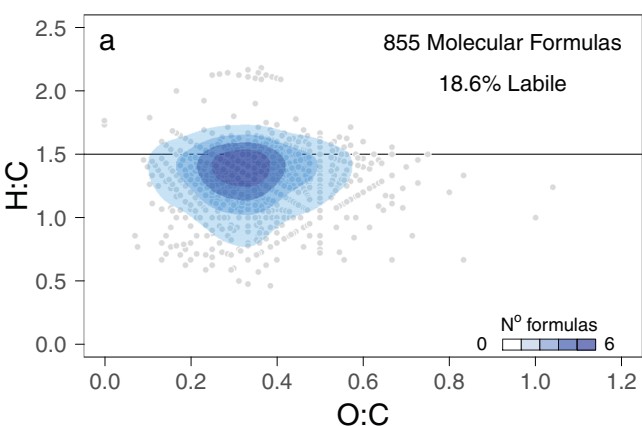

## Standard: Suwannee River Fulvic Acid

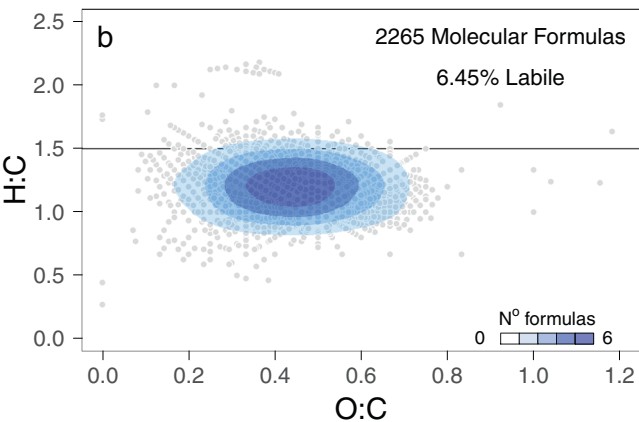

**Fig. 1 Plastics leach novel organic compounds with many more molecular formulae with a high index of lability.** We compared molecular formulae retrieved from FT-ICR-MS in (**a**) plastic leachate with (**b**) a freshwater standard sample widely used in mass spectrometry. Dots are individual molecular formula, with density representing the number of identical formulae along axes of H:C and O:C. Molecules were classed as having a high lability index based on a H:C ratio $\geq 1.5$ after D'Andrilli et al.[21].

production, such as isophthalic acid and phthalates, and 2 corresponded to known breakdown products unique to plastics (Table 1).

**Bacteria grow faster and more efficiently when offered a small amount of plastic leachate.** Plastic leachate increased both the BPP and BGE of natural bacterial communities after 3 days despite adding little carbon to lake DOM. The addition of plastic leachate increased mean [95% confidence interval, CI] BPP by 2.29 [1.92, 2.73] times compared to the control treatment (Fig. 2). Specifically, BPP increased from an estimated mean of 0.078 [0.058, 0.105] μg C L$^{-1}$ hr$^{-1}$ under the control treatment to 0.178 [0.132, 0.240] μg C L$^{-1}$ hr$^{-1}$ under the plastic treatment. We also found that bacterial growth was more efficient in the presence of plastic leachate than when only natural lake DOM was available. The addition of plastic leachate increased BGE by 1.72 [1.27, 2.32] times compared to the control treatment (Fig. 3a). Specifically, BGE increased from an estimated mean of 8.1 [5.8, 11.5] % in the control treatment to 14.0 [10.0, 19.5] % in the plastic treatment. To sustain a mean increase in BPP of 7.31 μg C L$^{-1}$ over the 72 h of our incubation with the estimated BGE of 14.0 [10.0, 19.5] %, bacteria would have to process a mean of 51.5 [37.0, 72.1] μg C L$^{-1}$, which is half that added by the leachate.

**Plastic leachate is used most efficiently in lakes with less diverse DOM.** Plastic leachate led to relatively greater increases in BGE in lakes with less functionally diverse DOM and less DOM itself (Fig. 3). We detected interactions between the plastic treatment and both lake FD and lake dissolved organic carbon (DOC) concentration (Fig. 3b, c). At a low FD, i.e. 1 standard deviation (SD) beneath the mean, bacteria were more efficient in the presence of plastics: BGE increased by a mean [95% CI] of 2.31 [1.54, 2.31] times from an estimated mean of 2.57 [1.71, 3.86] % to 5.93 [3.95, 8.89] %. In contrast, at a high FD (i.e. 1 SD above the mean), there was no change in BGE when plastic leachate was added: 1.18 [0.50, 2.80] times difference. BGE varied similarly with lake DOC concentration. At a low DOC concentration, BGE increased by 3.43 [2.82, 4.15] times from an estimated mean of 1.69 [1.39, 2.05] % to 5.77 [4.75, 6.99] %, whilst at a high DOC concentration there was no effect of plastic addition with a 0.74 [0.27, 2.04] times difference in BGE. Neither FD nor DOC influenced the extent to which BPP varied with plastic leachate, as the Akaike information criterion (AIC) increased by 1.52 and decreased only by 1.93, respectively, from retaining these treatment interactions during model selection. The absence of these interactions, despite influencing BGE, may ultimately reflect site-

**Table 1 Molecular formulas and putative compounds unique to plastic leachate DOM.**

| Molecular formula | Putative name | Main application | Abundance (%) |
|---|---|---|---|
| $C_{20}H_{26}O_4$ | dicyclohexyl phthalate | Hardener/plasticizer | 0.18 |
| $C_{14}H_{20}O_2$ | 2.6-di-tertbutyl-p-benzoquinone | Breakdown product | 0.01 |
| $C_{10}H_{14}O_2$ | 2-tert-butylhydroquinone | Rubber filler | 0.02 |
| $C_{10}H_{14}O_2$ | 4-tert-butylpyrocatechol | Solvent/adhesive | 0.02 |
| $C_{18}H_{24}O_6$ | butoxycarbonylmethyl butyl phthalate | Plasticizer | 0.05 |
| $C_{13}H_{10}O_5$ | 2.2'.4.4'-tetrahydroxybenzophenone | Antioxidant | 0.01 |
| $C_8H_8O_3$ | methyl 4-hydroxybenzoate | Antistatic/softener | 0.01 |
| $C_{18}H_{18}O_5$ | oxydiethylene dibenzoate | Plasticizer/softener | 0.01 |
| $C_8H_6O_4$ | isophthalic acid | Lubricant/adhesive | 0.17 |
| $C_{14}H_{12}O_3$ | oxybenzone | Stabilizer/lubricant | 0.01 |
| $C_{18}H_{26}O_4$ | dipentyl phthalate | Plasticizer | 0.04 |
| $C_{11}H_{16}O_2$ | tert-butyl-4-methoxyphenol | Antioxidant | 0.02 |
| $C_{17}H_{24}O_3$ | 7.9-di-tert-butyl-1-oxaspiro(4.5)deca-6.9-diene-2.8-dione | Breakdown product | 0.03 |

Isotope-free molecular formulas found exclusively in the leachate were cross-referenced against isotope-free molecular formulas databases of known plastic additives[14, 59]. Abundance was calculated relative to all 855 molecular formulas found in the leachate, of which 296 were unique, i.e. absent from the 22 lake samples after blank correction. Breakdown products were those derived from additives or other sources unique to plastic products.

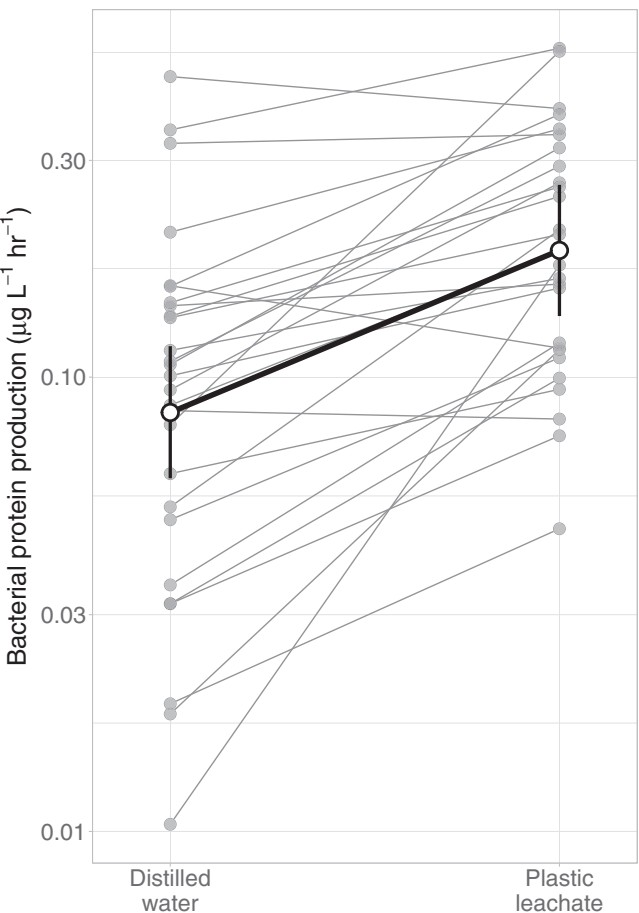

**Fig. 2 Carbon uptake measured as bacterial biomass production (BPP) increased with the addition of plastic leachate.** Bolded line shows the mean increase in BPP between treatment means ± 95% confidence intervals. Thin lines join mean effects for each of the 29 study lakes ($n = 3$ replicates per treatment per lake).

specific differences in the metabolic costs for bacteria to exploit available carbon (Supplementary Fig. 3). Other environmental variables retained as predictors of BGE during model selection, specifically water temperature, pH, and latitude, also did not influence the response of BGE to plastic leachate (ΔAIC from including interactions with leachate treatment: 0.24, 1.36, and 0.27, respectively).

**Bacterial diversity affects the efficiency of plastic leachate usage.** The effect of plastic leachate on BGE varied with the diversity of bacteria present in the lake, as expected if microbial community composition influenced the use of novel DOM sources. We retrieved 2148 amplicon sequence variants (ASVs) across 20 lakes subjected to 16S amplicon sequencing. Community composition was dominated by the genera *Acinetobacter*, *Exiguobacterium*, and *Brevundimonas* (Supplementary Fig. 4). We then summarised differences in bacterial diversity using the Shannon index, which ranged between 3.46 to 6.38 per lake, similar to other studies in northern waters[38]. We found that bacterial diversity interacted with the plastic treatment to influence BGE (Fig. 3d). At high bacterial diversity, plastic leachate addition increased BGE 2.93 [1.71, 5.03] times from an estimated mean of 6.59 [3.84, 11.3] % to 19.3 [11.2, 33.1] %. There was no effect of plastic addition at low bacterial diversity with a 1.08 [0.58, 1.99] times difference. Bacterial diversity also had no effect on the response of BPP to leachate addition, as expected if taxa

did not strongly discriminate in their use of labile, plastic-derived compounds, but instead used them with varying efficiency (ΔAIC from retaining interaction: 1.66).

To identify which genera responded most strongly to the plastic leachate, we tested if some ASVs were more abundant when BPP and BGE increased after leachate addition. We found that the fold increases in BPP and BGE were positively correlated with 154 and 540 ASVs, respectively (Supplementary Fig. 4). BPP and BGE increased most with the fold increase in ASVs that belonged to the genera *Hymenobacter* and *Deinococcus*, respectively (Supplementary Fig. 4).

## Discussion

Here we found that plastic-derived DOM was substantially different to natural DOM and that it strongly promoted bacterial growth. Plastic leachate more than doubled bacterial biomass production relative to the control treatment despite adding a mean (±SD) of only 4.5 ± 4.0% of the total lake DOC concentrations. As much of the carbon provided by the plastic leachate had to be assimilated to sustain the increase in BPP, given the mean BGE, this result further highlights the bioavailability of plastic leachate for use by microbial communities. Although the increase in BPP was less than the over 4-times increase reported in oceans by Romera-Castillo et al.[8], we added 7.4-times less DOC to replicate concentrations observed in lakes near population centres (Supplementary Methods 1). Therefore, we found strong effects of plastics at environmentally relevant concentrations, although differences between our study and others may be due to differences in the characteristics of background waters. These positive effects may disappear at higher leachate concentrations and/or in different waters, as found by Tetu et al.[11] who added 1.3- to 250-times more plastic-derived DOM than us into artificial seawater. By characterising the unique molecular properties of plastic leachate, our study now adds novel insight into why and when leachate stimulates bacterial growth. Specifically, the high lability and bioavailability of the plastic leachate likely increased BPP and BGE, as occurs with DOC in the natural environment[18,19,39,40]. An additional quantity of carbon from plastic leachate is unlikely to be the sole explanation for these results as it comprised only a small fraction of the total DOC pool. Our results also suggest that high plastic leachate concentrations, such as used by Tetu et al.[11], may impair bacterial growth because they add large quantities of toxic compounds, e.g. oxybenzone[41,42].

Increases in BGE varied with lake DOM concentration and composition, suggesting that the local environment mattered in addition to the leachate. As BGE, but not BPP, interacted with lake characteristics, local bacterial communities must have produced similar biomass at low and high FD/DOC concentrations but with lower metabolic costs in the former. Lower costs could arise because DOM contained proportionally more molecules with a high lability index at low FD/DOC concentrations once we added leachate to lake water (Supplementary Fig. 3). There may be lower metabolic costs when microbes have more labile substrates to consume, such as if it permits them to target molecules that are more thermodynamically available[43]. Microbial communities in these environments could have also specialised towards those that produce more efficient enzymes for degrading the available resources[30,44]. FD can reflect the number of niches available for microbial decomposers[20,23]. Therefore, bacteria in lakes with few niches (i.e. low FD) may benefit most from the high-lability-index molecules that we found were added by plastic leachate. Taken together, this dependency of bacteria on pre-existing DOM can explain why their responses have varied in studies of plastic leachate that have used different source

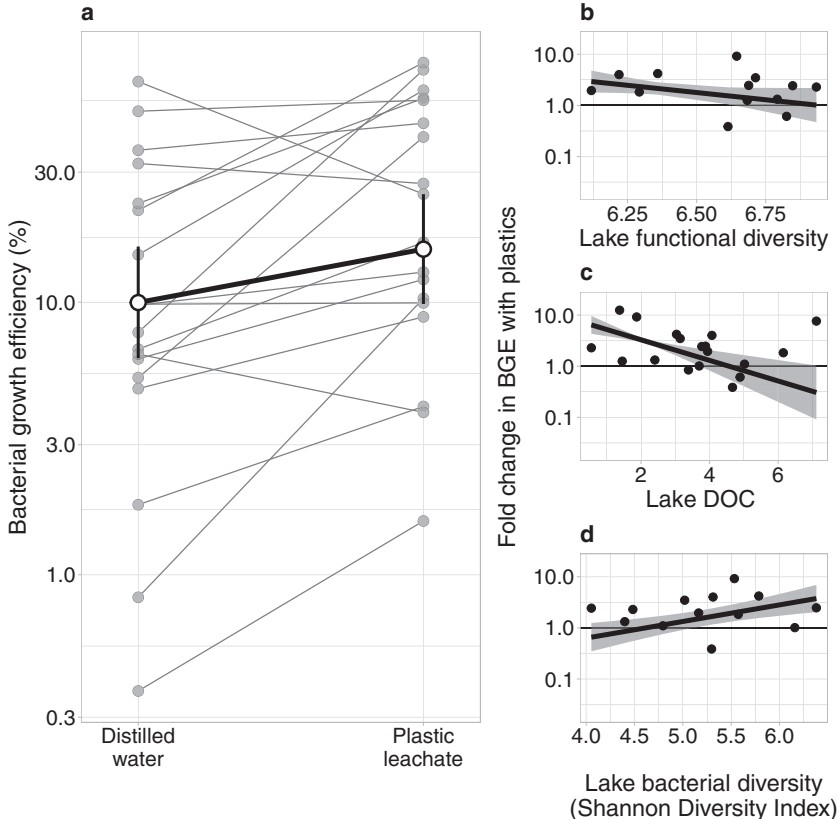

**Fig. 3 Bacterial growth efficiency (BGE) increased with the addition of plastic leachate depending on lake characteristics. a** Bolded line shows mean ± 95% CI for BGE in each treatment. Thin lines join mean effects for each of 18 study lakes with respiration data ($n = 1$ replicate per treatment per lake). BGE increased relatively less with plastic leachate addition as either the (**b**) functional diversity of lake dissolved organic matter (DOM) increased, (**c**) lake dissolved organic carbon (DOC) concentration increased, or (**d**) lake bacterial diversity decreased. Bolded lines are the estimated means ± 95% CIs for the trends and points are observed changes in BGE with plastic leachate addition. Horizontal line indicates no change in BGE with leachate addition (i.e. fold change = 1), whereas values above and below indicate an increase and decrease in BGE, respectively.

waters[8,11]. More generally, our results suggest that how microbes respond to plastic leachate depends on both the number of potential microbial niches conferred by existing DOM and the capacity of local communities to occupy these niches.

Microbial diversity also influenced the increase in BGE after leachate addition. We specifically found that increases in BGE after leachate addition were amplified at higher levels of bacterial diversity. Greater diversity may increase the likelihood of taxa that can use plastic-derived compounds efficiently, thereby elevating the BGE of the entire community. To our knowledge, no study has explored how bacterial diversity influences the extent to which BGE responds to resource manipulation. Previous studies have instead correlated BGE to bacterial richness[45,46], and changes in BGE to bacterial community composition[47]. More broadly, our results offer promise that some taxa may be particularly well suited to use plastic-derived compounds and remove them from the natural environment.

By providing an understanding of when plastic leachate is used by natural communities, our findings have wider implications for aquatic food webs and pollution mitigation efforts. First, more biomass at the base of the food webs will transfer more energy into higher trophic levels, stimulating the growth of higher organisms[48,49]. For example, *Daphnia* grew as quickly on microplastics as when fed algae[10], indicating that the increase in bacterial production from plastic-derived carbon can support the growth of higher trophic levels. Second, our results offer insight for efforts to identify environmental isolates that might remove plastic-derived compounds from the natural environment.

Specifically, we found ASVs in the genera *Deinococcus* and *Hymenobacter* were associated with high levels of plastic leachate use, consistent with previous observations of microbial communities associated with biodegradable plastic films[50]. *Deinococcus* taxa have also been shown to match DNA sequences encoding a recently identified polyethylene terephthalatase enzyme from *Ideonella sakaiensis*[51]. Other taxa positively correlated with bacterial metabolism included *Exiguobacterium*, which were previously found to grown solely on polystyrene film[52]. However, bacteria capable of utilising leachates may differ from those that degrade plastic itself. Recent studies have isolated phylogenetically divergent bacteria with the ability to degrade plastics, including strains of Proteobacteria—such as *Pseudomonas* spp.[53–55], Rhodobacteraceae[56], *Ideonella sakaiensis*[57], and *Acinetobacter baumannii*[58]—and Firmicutes such as *Bascillus* spp.[53,55]. Many of these taxa were strongly associated with BPP and BGE in our study (Supplementary Fig. 4). Irrespective of whether the microbes using leachates are the same as those decomposing it, the ability to uptake leachates is important for reducing chemical pollution from plastics[11,59] and our results identifying taxa that do so can help direct biological remediation efforts.

Our study has at least three limitations despite identifying clear effects of plastics on the metabolism of microbial communities. First, we focused solely on bacteria, but other microorganisms such as microalgae and fungi are also affected by plastics and plastic leachates[60–63]. These additional interactions may further influence the overall response of ecosystem metabolism to plastic

pollution in addition to the effects of bacteria observed here. Second, we only leached LDPE. The chemical composition of leachate from other plastics will likely differ and so the type of plastic present within lakes may also influence bacteria alongside the local environment. However, LDPE is the most common plastic found in aquatic systems[35], so should contribute most to the DOM pool available for use by bacteria. Finally, our study used a single LDPE concentration that was representative of plastic concentrations found in lakes near population centres (Supplementary Methods 1). Higher concentrations, such as found at waste management sites, may have less positive effects on microorganisms, especially if higher concentrations of toxic additives accumulate[11]. Irrespective, plastics will pollute the environment for decades[64]. Our findings are therefore valuable as they suggest that some lakes (e.g. high DOC concentrations, functionally diverse DOM, low bacterial diversity) are least able to remove leachate dissolved from plastics and so would benefit most from future pollution management.

## Methods

**Lake sampling**. We sampled 29 lakes across Scandinavia between August and September 2019. The lakes were located between latitudes of 59.1°N and 70.3°N to capture broad environmental gradients (Supplementary Fig. 1). For example, the lakes differed in depth (range: 0.9–303 m) and area ($0.01–464 \, km^2$), and, at the time of sampling, they differed in mean surface temperature (9.4–20.6 °C), pH (5.81–6.95), DOC concentration (0.55–7.97 mg $L^{-1}$), and DOM functional diversity (6.12–6.96).

Lakes were sampled at their deepest point. We collected 10 L of surface water in an acid washed Nalgene bottle. At 20 lakes, we immediately preserved microbial community composition by passing 1000 mL of water through a 0.2 µm Sterivex filter unit (Millipore). Filters were stored at −20 °C until laboratory analyses. We then measured the pH and temperature of lake water using a multiprobe (HI-99171, Hanna Instruments). Finally, total nitrogen (TN), DOC and DOM were sampled at 22 lakes by filtering 500 mL of water through pre-combusted glass fibre filters (0.5 µm nominal pore size, Macherey-Nagel) into three glass amber bottles with no headspace. Bottles were acidified to pH 2 with 0.5 mL of 10% HCl and stored in the dark. Remaining water was stored in the Nalgene sampling bottle for up to three hours in the dark before beginning the experiment.

**Plastic leachate preparation**. Plastic bags made of LDPE—the most common plastic type in freshwaters[35]—were collected from four major shopping chains (John Lewis, Superdrug, Clintons, and Next) in Cambridge, England and cut into $1 \, cm^2$ squares. 240 squares (60 from each shopping chain) were incubated in 150 mL of distilled water at 25 °C for 7 days under an LED lamp that simulated natural UV exposure (395–530 nm wavelength, 100 µmol photons $m^{-2} \, s^{-1}$ light intensity) and with constant agitation to simulate environmental transport[8]. A separate flask of 125 mL of distilled water without the plastics was also incubated under the same conditions as a control that confirmed no DOM was leached from our treatment process. At the end of the incubation, water was filtered for use in the experiment through pre-rinsed 0.2 µm cellulose acetate syringe filters (Sartorius AG) into dark, pre-combusted glass vials with no headspace. We used a more restrictive filter size than when preserving lake DOM as we wanted to ensure absolutely no lab microbes could contaminate the experimental treatments and be introduced into lake waters. The incubations were preserved for DOM and DOC measurements as for lake waters. We used water from the 0.2 µm filtrate rather than a higher pore size like in the lakes to measure precisely what was added in the experimental treatments.

**DOM characterisation**. We estimated the functional diversity (FD) of lake water and plastic leachate DOM using Fourier-transform ion cyclotron resonance mass spectrometry (FT-ICR-MS). The DOM was solid phase extracted as previously described in Dittmar et al.[22]. Briefly, the DOM from the 500 mL bottles was retained on 1 g of a styrene-divinylbenzene polymer (Bond Elut PPL, Agilent) and eluted with 4 mL of ultrapure methanol (LC-MS LiChrosolv, Merk). The resulting extracts were diluted in a 1:1 (v:v) methanol:water solution to a final concentration of 2.5 ppm. 100 µL of the diluted extracts were directly infused in negative mode via electrospray ionisation into a 15 Tesla Solarix XR FT-ICR-MS (Bruker Daltonics, Germany). 200 scans were collected for each lake, and the scans were then calibrated using DataAnalysis software (Bruker Daltonics, Germany). Masses in the range 150 to 1000 m/z were exported and the online platform ICBM-OCEAN[65] used to assign molecular formulas. FD was computed as in Mentges et al.[23], using differences in the number of carbon atoms in the molecular formulas, whereby a greater value indicated more diversity in the size of the molecular formulas. We also estimated the bioavailability of the plastic leachate and the lake water DOM by classifying molecular formulas with a H:C ratio ≥1.5 as having a high lability

index[21]. DOC and TN concentrations in the samples were measured within a month of sampling on a Shimadzu TOC-L TNM-L analyser (Shimadzu Corporation, Japan).

**Experimental design**. At each lake, incubations were set up to test the effect of plastic leachate (Supplementary Fig. 2). Nine 125 mL glass bottles were filled with 125 mL of the collected lake water. Three bottles received either 4.6 mL of leachate, 4.6 mL of distilled water, or no further addition. The volume of leachate was determined so that 0.1 mg C $L^{-1}$ was added. This concentration was assumed to be representative of the amount of carbon leached from plastics in the environment based on: (1) the concentration of plastics in lakes near cities in southern Europe, (2) the density and volume of LDPE plastic bags, and (3) the expected leaching rate of plastics (calculations in Supplementary Methods 1). Bottles were crimped airtight with PTFE/rubber septa, ensuring that there were no bubbles present before proceeding. Pure lake water bottles were processed directly to provide measurements for the start of the incubation, whilst bottles that received the distilled water or plastic leachate addition were incubated for 72 h in the dark at ambient temperature. Identical vials were also prepared for oxygen concentration measurements to derive BGE. Lake water with either plastic leachate or lake water with distilled water—as previously described—were added to gastight 25 mL glass vials in triplicate with no headspace. Plastic leachate or distilled water (0.9 mL) was added to the same concentration (0.1 mg C $L^{-1}$) as the incubation described above.

**Bacterial activity**. To determine bacterial activity, BPP and respiration were measured after a 72-h incubation. Bacterial productivity was estimated based on protein production using carbon uptake as a proxy[36]. Briefly, 17 nM of [$^3$H]-leucine was added to 1.5 mL of sample water collected from each incubation bottle into a 2 mL centrifuge tube. 300 µL of 50% trichloroacetic acid (TCA) was then added to one sample from each treatment in each lake (hereafter referred to as "killed") with nothing added to the other sample (hereafter referred to as "live"). All samples were incubated in the dark at lake temperature for 1 h. At the end of the incubation, 300 µL of 50% TCA was added to the live samples. Cells were precipitated by centrifugation (10 min, 16,000 × g). Pellets were washed with 1 mL of 5% TCA, centrifuged again (10 min, 16,000 × g), and the supernatant removed. Samples were air dried before adding 1 mL of Optiphase HiSafe 3 liquid scintillation cocktail. Counts per minutes (CPM) were measured using a Triathler liquid scintillation counter (Hidex Oy, Finland), alongside a standard of known concentration and two blanks (1 mL of scintillation fluid only, and an empty Eppendorf tube) used for calibration. CPMs were converted to disintegrations per minute, subtracting the killed and blank from each live value, and adjusting for counting efficiency based on the standard. These values were then converted to carbon uptake[66].

Oxygen levels in the water were measured before and after the incubation to determine respiration rate. One vial from each treatment was measured immediately, and the other two were measured after 72 h in the dark. We used fibre-optics optodes connected to a OXY-1 ST metre (PreSens, Germany) to record oxygen concentration as percentage of air saturation in each 25 mL vial[67,68]. Readings were registered every second until a steady state had been reached—for 90% of samples this was reached within 5 min. Oxygen concentration was then derived from the median of the last 10 stable values in the time series. Pressure, temperature, and salinity were also recorded and used to correct the values to standard conditions.

To determine whether bacteria used carbon efficiently for growth, bacterial growth efficiency (BGE) was calculated as reviewed by del Giorgio and Cole[69]. BPP and respiration were converted to units of moles of carbon per hour, assuming a respiratory quotient of one, and we then calculated the proportion of total carbon incorporated into biomass by dividing the carbon used for growth (BPP) by the sum of BPP and respiration.

**Bacterial community composition**. In addition to DOM characteristics, we considered how the composition and diversity of bacteria influenced their responses to plastic leachate. To characterise bacterial communities, DNA was extracted from the Sterivex filters following an established protocol[70] with minor modifications.

Briefly, we placed the filters, which were separated from the filtration unit under sterile conditions, into a cryotube containing silica and zirconia beads (3.0, 0.7, and 0.1 mm diameter) before vortexing at 2850 rpm for 15 min. Then, we added 0.6 mL of phenol-chloroform-isopropanol (25:24:1), 0.6 mL of 5% cetrimonium bromide, 60 µl of 10% sodium dodecyl sulfate, and 60 µl 10% N-lauroylsarcosine and vortexed the solution at 2850 rpm for 15 min. We then centrifuged the samples at 16 × g for 15 min at 4 °C and collected the supernatant. To the supernatant, we added an equal volume (ca. 0.6 mL) of chloroform-isopropanol (24:1), mixed the samples by inversion, and centrifuged at 16 × g for 10 min at 4 °C. We again collected the supernatant and precipitated the DNA at 4 °C overnight in polyethylene glycol with 1.6 M sodium chlorine. We centrifuged the samples again at 17 × g for 90 min at 4 °C, removed the supernatant, and washed the pellet with ice-cold (−20 °C) 70% ethanol. The DNA was dissolved in ultrapure water and quantified on a Qubit fluorometer (ThermoFisher, USA). We also extracted DNA from a ZymoBIOMICS™ Microbial Community Standard (Zymo Research, USA)

and nuclease-free water (Qiagen, Germany) to act as positive and negative controls, respectively. Libraries were prepared exactly like the lake samples.

We amplified the V6 and V8 regions of the 16S rRNA gene using the bacteria specific primers[71] 5' ACGCGHNRAACCTTACC 3' and 5' ACGGGCRGTGWG TRCAA 3'. Samples were sequenced at $2 \times 300$ bp paired-end on an Illumina MiSeq at the Integrated Microbiome Resource (Halifax, Nova Scotia, Canada)[71]. No DNA was retrieved from the negative control and no contaminants were present in the positive control. We then removed the primers from the raw sequences using *cutadapt*[72] and assigned taxonomy with the *DADA2* pipeline[73] and the Silva v132 database[74]. Overall, 1.7 million reads were classified into 2148 amplicon sequence variants (ASVs), which represented 75% of the total raw reads, and we used these to compute the Shannon diversity index[75]. The raw sequences have been deposited in the EBI database under accession number PRJEB49321.

**Statistical analysis**. The effect of plastics on BPP and BGE were tested using linear mixed effects models. As both BPP and BGE were not normally distributed, they were natural log transformed before analysis. We then considered the following fixed predictors for each bacterial response: functional diversity of lake DOM, bacterial diversity (Shannon index), DOC and TN concentrations, lake water temperature, pH, and latitude. The latter variable was included to control for differences in lake location, which is known to influence bacterial community composition[76], and so which we hypothesised may affect the overall bacterial response. We included an interaction in our model between each predictor and the experimental treatment (i.e. plastic or control treatment), which was also included as a main effect. We accounted for repeated measurements of the same lake by including lake ID as a random effect. Models were initially fitted using maximum-likelihood with the *lmer* function from the *lme4* package in R version 3.5.3[77]. To avoid multicollinearity, we inspected correlations among model parameter estimates. When two variables were correlated with a Pearson correlation $r > 0.90$, the most biologically relevant term was selected for inclusion into the model. The best supported model was then determined using backwards stepwise elimination using the *drop1* function from *lme4*. Fixed effects were dropped if their retention would not have decreased the model's Akaike information criterion score by more than two. Only results from the best supported model, re-fitted using restricted maximum likelihood, were reported in the main text. Confidence intervals were calculated from these models using the *emmeans* package[78].

We identified which ASVs were associated with changes in BPP and BGE after plastic leachate addition. We separately estimated the log2-fold change in the relative abundance of each ASV relative to fold-increases in BGE or BPP by fitting separate negative binomial generalised linear models to read counts using the *DESeq* function in the *DESeq2*[79] R package. All ASVs with <100 reads were removed to avoid inferring correlations with rare taxa that may be subject to more stochastic variation in abundance. *P* values were adjusted to correct for multiple comparisons with the Benjamini–Hochberg method[79] and considered statistically significant beneath a threshold of 0.05.

**Reporting summary**. Further information on research design is available in the Nature Research Reporting Summary linked to this article.

## Data availability

The BPP and BGE data generated in this study can be downloaded from FigShare (https://figshare.com/articles/dataset/BPP_data/19692031; https://figshare.com/articles/dataset/BGE_data/19692028). The DNA sequences can be downloaded from the EBI database (https://www.ebi.ac.uk/services/dna-rna) under accession number PRJEB49321.

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

## Acknowledgements

We thank Carolyn Ewins, Sophie Guillaume, and Sam Woodman for help with field work. This work was funded by a H2020 ERC Starting Grant 804673 to AJT.

## Author contributions

E.A.S., J.A.F., D.C.A, and A.J.T. designed the study. E.A.S., J.A.F., S.C., and Y.Z. performed experimental work. E.A.S., J.A.F., T.D., and A.J.T. analysed data. T.D., D.C.A., and A.J.T. supervised the study. E.A.S., J.A.F., and A.J.T. wrote the first draft of the manuscript and all co-authors commented on and approved the final manuscript.

## Competing interests

The authors declare no competing interests.
