## [Peer Review File · Nature Communications]

Plastic pollution fosters more microbial growth in lakes than natural organic matterREVIEWER COMMENTS

Reviewer #1 (Remarks to the Author):

The manuscript reports a series of experiments aimed to evaluate the impact of plastic leachate in the bacteria functioning (DOM use) in several lakes in Scandinavia.

The work is overall very well performed and described in the article. The topic of the work is of interest for the audience of the journal, since it tackles the impact of plastic contamination in the natural aquatic environment. Although interesting there are some aspects of the work that need to be clarified before considering it for publication.

The hypothesis of the work is that natural DOM characteristics are impacting how the bacterial community would use plastic leachate. But what about other environmental factors in each of the lakes? What about Temperature in each of the lakes? What about nutrients or pollution status? What about the overall microbial composition and diversity? I think that not only DOM but many other environmental parameters should be taken into account to evaluate bacterial response to plastic leachate. Actually, I think that microbial diversity cannot be disregarded.

The abstract is not clear enough in my opinion. 2 main points are not clear to me:

When plastic leachate was added, bacterial biomass acquisition and bacterial growth were respectively 2 and 2.2. times higher than control samples. Is the difference between these 2 ratios relevant enough (statistically) to make any conclusion? (in this case conclusion is about the bioavailability of plastic leachate in comparison to natural DOM).

Moreover, it is not that obvious why decoupling of bacterial growth and biomass acquisition is related to the bioavailability of organic matter. The authors try to explain that hypothesis in the introduction (lines 89-92 but it is not clear enough.

Moreover, I think that numbers and ratios should be clearer in the abstract: based on the values indicated in lines 143 and 148 the increase ratio of BPP and BGE were 2.29 and 2.16 respectively, which are quite similar. Please give these 2 values in the abstract.

I suggest a reformulation of the abstract to give a clearer message

Line 63-64. Please reformulate the sentence to clarify the topic or limitations of the 3 articles referred to better stress the novelty of the present work.

Line 68. The authors are focusing on bacteria to investigate the plastic leachate deployment, but what about other microorganisms in the lake water column? What about microalgae for instance? Are not they able to assimilate plastic leachate?

Line 92-95. This sentence is rather placed at the conclusions section than in the introduction

Line 102. I think it is worth indicating what FD stands for

Line 99. Based on what is presented-discussed in this section I would rather say that plastic leachate is more labile than natural organic matter. Otherwise, add any explanation in the section about bioavailability

Line 269. Simulated natural? UV exposure?

Line 273. Why was water filtered here with different filters than in line 260?

Line 278. Was plastic leachate DOM not characterized as well?

Reviewer #2 (Remarks to the Author):

The study described in the manuscript makes a variable contribution to our understanding of the impact of plastic pollution on microbial growth in the aquatic environment. The study was well designed and the interpretations sound. Your manuscript is therefore very good.

However, there are two limitations, one of which I think you raised yourself in the discussion. That is the use of leachate from only LDPE, this of course as you are well aware could limit your findings, like other types of plastics may have a different impact on bacterial growth.

Secondly, you only used one concentration of the LDPE leachate in your study, although you compared it to other concentrations used elsewhere. However, as you have acknowledged yourself, the water characteristics could have an impact on the utility of the leachate by the bacteria. Therefore, relying on findings from other studies to make conclusions on the impact of concentration of the leachate on bacterial growth is also problematic.

The last concern I have is in relation to the non-specificity of the bacteria in your study. As you may be well aware, the growth of bacteria in response to different carbon sources could differ. Therefore, you may find that the leachate could potentially have different impacts on different bacterial communities. Therefore, making a conclusion on the impact of LDPE leachate on bacterial growth, in general, is also problematic without first ascertain if the impact is the same for different groups of bacteria.

Based on these concerns, I suggest you add a study limitation section, where you discuss these limitations clearly.

Reviewer #3 (Remarks to the Author):

Potential negative impact of plastic pollution for ocean and freshwaters has been under discuss for last five to ten years. Meanwhile few studies (Yoshida et al. 2016, Taipale et al. 2019) has directly studied composition of (micro)plastics and found it to be relatively slow process, authors here study first time the impact of leachate from plastic bags. Unlike plastic decomposition you have found that leachates from plastic bags are readily usable and can support microbial activity. Moreover, you have found that leachates from plastic bags are more readily usable than DOM. Your incubation studies cover 29 lakes from Scandinavia and experiments are well planned and done. I keep that your finding will be important addition to plastic pollution literature. My only suggestion for you is that you should more clearly separate plastic decomposition and utilization of leachates from plastic bags. Meanwhile leachates from plastic bags can be formed through abiotic process without microbial activity, plastic decomposition itself means microbial decomposition of plastic. These two processes are separate and meanwhile plastic itself is recalcitrant for microbial degradation, leachates from plastics can be labile and easily utilized by microbes. Therefore, microbes benefitting from leachates differs most likely those bacterial taxa decomposing plastic itself. Clear water lakes usually contain more labile organic carbon than humic waters, and thus microbes in humic waters can decompose complex organic compounds. You have reported here higher respond of microbes in low DOC than high DOC lake, which supports that leachates from plastic bags are easily degradable unlike plastic itself. This should be clearly mentioned in the abstract and debated in the discussion. Please, note also that microbes utilizing leachates are not necessarily same as plastic decomposers. Therefore I ask you kindly rephrase discussion that readers would not mix these two things.

Reviewer #1 (Remarks to the Author):

The manuscript reports a series of experiments aimed to evaluate the impact of plastic leachate in the bacteria functioning (DOM use) in several lakes in Scandinavia.

The work is overall very well performed and described in the article. The topic of the work is of interest for the audience of the journal, since it tackles the impact of plastic contamination in the natural aquatic environment. Although interesting there are some aspects of the work that need to be clarified before considering it for publication.

We thank the Reviewer for their positive feedback here and address their concerns below.

The hypothesis of the work is that natural DOM characteristics are impacting how the bacterial community would use plastic leachate. But what about other environmental factors in each of the lakes? What about Temperature in each of the lakes? What about nutrients or pollution status? What about the overall microbial composition and diversity? I think that not only DOM but many other environmental parameters should be taken into account to evaluate bacterial response to plastic leachate. Actually, I think that microbial diversity cannot be disregarded.

We agree that other environmental parameters, like water temperature and nutrients, and microbial diversity may be important for the response of bacteria to plastic leachate. We have now included these data as predictors in our statistical models of BPP and BGE. These changes only further strengthen our results, so we thank the Reviewer for their suggestion.

Added on lines 305-312:

“Lakes were sampled at their deepest point. We collected 10 L of surface water in an acid washed Nalgene bottle. At 20 lakes, we immediately preserved microbial community composition by passing 1000 mL of water through a 0.2 µm Sterivex filter unit (Millipore). Filters were stored at -20°C until laboratory analyses. We then measured the pH and temperature of lake water using a multiprobe (HI-99171, Hanna Instruments). Finally, total nitrogen (TN), DOC and DOM were sampled at 22 lakes by filtering 500 mL of water through pre-combusted glass fiber filters (0.5µm nominal pore size, Macherey-Nagel) into three glass amber bottles with no headspace.”

Added on lines 402-429:

“In addition to DOM characteristics, we considered how the composition and diversity of bacteria influenced their responses to plastic leachate. To characterise bacterial communities, DNA was extracted from the Sterivex filters following an established protocol⁷⁰ with minor modifications. Briefly, we placed the filters, which were separated from the glass filtration unit under sterile conditions, into a cryotube containing silica and zirconia beads (3.0, 0.7, and 0.1 mm diameter) before vortexing at 2,850 rpm for 15 min. Then, we added 0.6 mL of phenol-chloroform-isopropanol (25:24:1), 0.6 mL of 5% cetrimonium bromide, 60 µl of 10% sodium dodecyl sulfate, and 60 µl 10% N-lauroylsarcosine and vortexed the solution at 2,850 rpm for 15 min. We then centrifuged the samples at 16 g for 15 min at 4°C and collected the supernatant. To the supernatant, we added an equal volume (ca. 0.6 mL) of chloroform-isopropanol (24:1), mixed the samples by inversion, and centrifuged at 16 g for 10 min at 4°C. We again collected the supernatant and precipitated the DNA at 4°C overnight in polyethylene glycol with 1.6M sodium chloride. We centrifuged the samples again at 17 g for 90 min at 4°C, removed the supernatant, and washed the pellet with ice-cold (-20°C) 70% ethanol. The DNA was dissolved in ultrapure water and quantified on a Qubit fluorometer (ThermoFisher, USA). We also extracted DNA from a ZymoBIOMICS™ Microbial Community Standard II (Log Distribution) (Zymo Research, USA) and nuclease-free water

(Qiagen, Germany) to act as positive and negative controls, respectively. Libraries were prepared exactly like the lake samples.

We amplified the V6 and V8 regions of the 16S rRNA gene using the bacteria specific primers⁷¹ 5' ACGCGHNRAACCTTACC 3' and 5' ACGGGCRGTGWGTRCAA 3'. Samples were sequenced at 2×300 bp paired-end on an on an Illumina MiSeq at the Integrated Microbiome Resource (Halifax, Nova Scotia, Canada)⁷¹. No DNA was retrieved from the negative control and no contaminants were present in the positive control. We then removed the primers from the raw sequences using *cutadapt*⁷² and assigned taxonomy with the *DADA2* pipeline⁷³ and the Silva v132 database⁷⁴. Overall, 1.7 million reads were classified into 2,148 amplicon sequence variants (ASVs), which represented 75% of the total raw reads, and we used these to compute the Shannon diversity index⁷⁵. The raw sequences have been deposited in the EBI database under accession number PRJEB49321.”

Added on lines 432-438:

“The effect of plastics on BPP and BGE were tested using linear mixed effects models. As both BPP and BGE were not normally distributed, they were natural log transformed before analysis. We then considered the following fixed predictors for each bacterial response: functional diversity of lake DOM, **bacterial diversity (Shannon index)**, DOC and **TN concentrations, lake water temperature, pH**, and latitude. The latter variable was included to control for differences in lake location, which is known to influence bacterial community composition⁷⁶, and so which we hypothesized may affect the overall bacterial response.”

We have added new paragraphs in the Results and Discussion explaining the effect of bacterial diversity and a panel to the BGE figure (Figure 3) to visualise these effects.

Added on lines 172-186:

“Bacterial diversity affects the efficiency of plastic leachate usage

The effect of plastic leachate on BGE varied with the diversity of bacteria present in the lake, as expected if microbial community composition influenced the use of novel DOM sources. We retrieved 2,148 amplicon sequence variants (ASVs) across 20 lakes subjected to 16S amplicon sequencing. Community composition was dominated by the genera *Acinetobacter*, *Exiguobacterium*, and *Brevundimonas* (Fig. S4). We then summarised differences in bacterial diversity using the Shannon index, which ranged between 3.46 to 6.38 per lake, similar to other studies in northern waters³⁹. We found that bacterial diversity interacted with the plastic treatment to influence BGE (Fig. 3d). At high bacterial diversity, plastic leachate addition increased BGE 2.93 [1.71, 5.03] times from an estimated mean of 6.59 [3.84, 11.3] % to 19.3 [11.2, 33.1] %. There was no effect of plastic addition at low bacterial diversity with a 1.08 [0.58, 1.99] times difference. Bacterial diversity also had no effect on the response of BPP to leachate addition, as expected if taxa did not strongly discriminate in their use of labile, plastic-derived compounds, but instead used them with varying efficiency (Δ AIC from retaining interaction: 1.66).”

Added on lines 200-208:

“Figure 3. Bacterial growth efficiency (BGE) increased with the addition of plastic leachate depending on lake characteristics. (a) Bolded line shows mean \pm 95% CI for BGE in each treatment. Thin lines join mean effects for each of 18 study lakes with respiration data (n = 1 replicate per treatment per lake). BGE increased relatively less with plastic leachate addition as either the **(b)** functional diversity of lake DOM increased, **(c)** lake DOC concentration increased, or **(d)** lake bacterial diversity decreased.”

Added on lines 249-257:

“Microbial diversity also influenced the increase in BGE after leachate addition. We specifically found that increases in BGE after leachate addition were amplified at higher levels of bacterial diversity. Greater diversity may increase the likelihood of taxa that can use plastic-derived compounds efficiently, thereby elevating the BGE of the entire community. To our knowledge, no study has explored how bacterial diversity influences the extent to which BGE responds to resource manipulation. Previous studies have instead correlated BGE to bacterial richness^{46,47}, and changes in BGE to bacterial community composition⁴⁸. More broadly, our results offer promise that some taxa may be particularly well suited to use plastic-derived compounds and remove them from the natural environment.”

We have also mentioned this result in the Abstract.

Added on lines 25-27:

“These effects varied with both the availability of alternate, especially labile, carbon sources and **bacterial diversity**.”

Additionally, we have explored the effects of microbial diversity in greater detail by associating individual taxa with the changes in BPP and BGE in response to Reviewer #2.

Added on lines 451-458:

“We identified which ASVs were associated with changes in BPP and BGE after plastic leachate addition. We separately estimated the log₂-fold change in the relative abundance of each ASV relative to fold-increases in BGE or BPP by fitting separate negative binomial generalised linear models to read counts using the *DESeq* function in the *DESeq2*⁷⁹ R package. All ASVs with <100 reads were removed to avoid inferring correlations with rare taxa that may be subject to more stochastic variation in abundance. P-values were adjusted to correct for multiple comparisons with the Benjamini-Hochberg method⁷⁹ and considered statistically significant beneath a threshold of 0.05.”

Added on lines 187-191:

“To identify which genera responded most strongly to the plastic leachate, we tested if some ASVs were more abundant when BPP and BGE increased after leachate addition. We found that the fold increases in BPP and BGE were positively correlated with 154 and 540 ASVs, respectively (Fig S4). BPP and BGE increased most with the fold increase in ASVs that belonged to the genera *Hymenobacter* and *Deinococcus*, respectively (Fig. S4).”

Added on lines 57-68 of Supplementary Materials:

“**Figure S4. Use of plastic leachate positively correlates with abundances of different microbial taxa.** Each dot is the mean log₂-fold change in normalized abundance of a given genus with increasing fold change in (a) BGE or (b) BPP. Responses of individual ASVs were averaged at the genus-level, n = 540 and 154 ASVs for (a) and (b), respectively. (c) Mean relative abundance (percent of total reads) in each lake for the 20 most abundant genera. For all, error bars are the standard deviation when more than one ASV was included in a genus. Underlined genera were always associated with community-level metabolism, that is, occur in (a) and (b). Bold genera were the most abundant and so occurred in (c) and either (a) or (b). Colors denote phyla: Actinobacteriota (red), Bacteroidota (blue), Bdellovibrionota (orange), Cyanobacteria (green), Deinococcota (brown), Firmicutes (yellow), Proteobacteria

(black), and Verrucomicrobiota (gold). ANPR = *Allorhizobium-Neorhizobium-Pararhizobium-Rhizobium*.”

Finally, the Reviewer raises the question of plastic pollution, but all lakes were contaminated and concentrations were measured on too small a subset of lakes to include formally in the statistical analyses. For these reasons, we simply mention this point elsewhere:

Added on lines 115-117:

“Of our study lakes, we previously surveyed 19 for pollution impacts and all were contaminated with microplastics and anthropogenic fibres³⁷. Thus, it is likely we detected associated plastic-derived compounds in DOM of these lakes.”

The abstract is not clear enough in my opinion. 2 main points are not clear to me: When plastic leachate was added, bacterial biomass acquisition and bacterial growth were respectively 2 and 2.2. times higher than control samples. Is the difference between these 2 ratios relevant enough (statistically) to make any conclusion? (in this case conclusion is about the bioavailability of plastic leachate in comparison to natural DOM).

If we understand the comment correctly, the Reviewer is asking if the ratios between the control and treated samples differed between bacteria biomass acquisition and bacteria growth efficiency. We did not intend to compare these two ratios, but can appreciate from rereading the text why the Reviewer may have gotten this impression. We simply want to say that bacteria biomass acquisition increased when we added carbon from plastic leachate. The accompanying increase in growth efficiency suggests that this was because bacteria could grow more efficiently rather than simply because they had access to more carbon. We have revised the Abstract accordingly.

Added on lines 23-25:

“These results were not solely attributable to the amount of dissolved organic carbon (DOC) provided by the leachate. Bacterial growth was also 1.72-times more efficient with plastic leachate **because the added carbon was more accessible than natural organic matter.**”

Moreover, it is not that obvious why decoupling of bacterial growth and biomass acquisition is related to the bioavailability of organic matter. The authors try to explain that hypothesis in the introduction (lines 89-92 but it is not clear enough.

Changed on lines 82-88:

“BGE allows us to separate whether BPP increases with leachate simply because more carbon is available or because the added carbon is also more labile and thus more accessible to bacteria. In the former case, BGE would remain unchanged, as any increase in BPP would purely be the result of an increase in the absolute amount of carbon processed rather than any change in how it was processed. In the latter case, BGE would increase along with BPP because the carbon would be processed more efficiently, such as if it was more bioavailable to resident bacterial communities.”

Moreover, I think that numbers and ratios should be clearer in the abstract: based on the values indicated in lines 143 and 148 the increase ratio of BPP and BGE were 2.29 and 2.16 respectively, which are quite similar. Please give these 2 values in the abstract.

We have now given these values in the Abstract. Because of reanalysing our data with bacterial diversity and the new environmental variables as predictors, these numbers are slightly different to the first version and are now 2.29 and 1.72, respectively.

Changed on lines 21-25:

“Consequently, plastic leachate **increased** bacterial biomass acquisition **by 2.29-times** when added at an environmentally-relevant concentration to lake surface waters. These results were not solely attributable to the amount of dissolved organic carbon (DOC) provided by the leachate. Bacterial growth was **1.72-times** more efficient with plastic leachate...”

I suggest a reformulation of the abstract to give a clearer message

We hope that the changes we have made to the Abstract detailed above in response to the Reviewer’s point-by-point comments have now made it give a clearer message.

Line 63-64. Please reformulate the sentence to clarify the topic or limitations of the 3 articles referred to better stress the novelty of the present work.

Changed on lines 55-57:

“Previous studies^{8,9,11} have shown how the response of bacteria to plastic leachate can vary, but, to our knowledge, no study has tested whether the molecular composition of DOM may explain this variation.”

Line 68. The authors are focusing on bacteria to investigate the plastic leachate deployment, but what about other microorganisms in the lake water column? What about microalgae for instance? Are not they able to assimilate plastic leachate?

We thank the Reviewer for this comment and agree that other microorganisms may have important interactions with plastic leachate. Microalgae are indeed affected by plastic leachate (reviewed by Prata et al. 2019, <https://doi.org/10.1016/j.scitotenv.2019.02.132>), with some papers focussing on plastic leachate rather than solid microplastics including: Schiavo et al. 2021, <https://doi.org/10.1080/15287394.2020.1860173>; Luo et al. 2019, <https://doi.org/10.1016/j.scitotenv.2019.04.401>. We have now added this point about other microorganisms in the Discussion, where we discuss our study more widely.

Added on lines 281-284:

“First, we focused solely on bacteria, but other microorganisms such as microalgae and fungi are also affected by plastics and plastic leachates⁶⁰⁻⁶³. These additional interactions may further influence the overall response of ecosystem metabolism to plastic pollution in addition to the effects of bacteria observed here.”

Line 92-95. This sentence is rather placed at the conclusions section than in the introduction. As suggested, we have incorporated this sentence into the summary paragraph of the Discussion.

Added on lines 223-225

“By characterising the unique molecular properties of plastic leachate, our study now adds novel insight into why and when leachate stimulates bacterial growth.”

Line 102. I think it is worth indicating what FD stands for

Thanks for spotting that we did not define this here.

Added on lines 98-99:

“We used a widespread **functional diversity (FD)** index to calculate ...”

Line 99. Based on what is presented-discussed in this section I would rather say that plastic leachate is more labile than natural organic matter. Otherwise, add any explanation in the section about bioavailability

We agree and have changed all instances of “bioavailable” in this section to “labile”.

Line 269. Simulated natural? UV exposure?

We have added the word “natural” and agree that it improves the description.

Added on lines 320-321:

“simulated **natural** UV exposure”

Line 273. Why was water filtered here with different filters than in line 260?

We wanted to be extra cautious with this step to ensure that no microbes could be filtered from the lab waters into the leachate and then introduced into the lake waters.

Changed on lines 326-331:

“We used a more restrictive filter size than when preserving lake DOM as we wanted to ensure absolutely no lab microbes could contaminate the experimental treatments and be introduced into lake waters. The incubations were preserved for DOM and DOC measurements as for lake waters. We used water from the 0.2 μm filtrate rather than a higher pore size like in the lakes to measure precisely what was added in the experimental treatments.”

Line 278. Was plastic leachate DOM not characterized as well?

Plastic leachate DOM was characterised and we thank the Reviewer for noticing this typo.

Added on lines 334-335:

“We estimated the functional diversity (FD) of lake water **and plastic leachate** DOM using Fourier transform ion cyclotron resonance mass spectrometry (FT-ICR-MS).”

Reviewer #2 (Remarks to the Author):

The study described in the manuscript makes a variable contribution to our understanding of the impact of plastic pollution on microbial growth in the aquatic environment. The study was well designed and the interpretations sound. Your manuscript is therefore very good.

We thank the Reviewer for this positive feedback on our study.

However, there are two limitations, one of which I think you raised yourself in the discussion. That is the use of leachate from only LDPE, this of course as you are well aware could limit your findings, like other types of plastics may have a different impact on bacterial growth.

We acknowledge this concern of the Reviewer and have addressed it further in a new paragraph focused on study limitations. We would also like to remind the reviewer that we work with LDPE because it is the most abundant plastic in freshwater (Schwarz et al. 2019; <https://doi.org/10.1016/j.marpolbul.2019.04.029>). Moreover, a study published by our group on 19 of the same lakes as in this study (Tanentzap et al. 2021; <https://doi.org/10.1371/journal.pbio.3001389>) also found LDPE to be the most common synthetic polymer.

Added on lines 285-288:

“Second, we only leached LDPE. The chemical composition of leachate from other plastics will likely differ and so the type of plastic present within lakes may also influence bacteria alongside the local environment. However, LDPE is the most common plastic found in aquatic systems³⁵, so should contribute most to the DOM pool available for use by bacteria.”

Secondly, you only used one concentration of the LDPE leachate in your study, although you compared it to other concentrations used elsewhere. However, as you have acknowledged yourself, the water characteristics could have an impact on the utility of the leachate by the bacteria. Therefore, relying on findings from other studies to make conclusions on the impact of concentration of the leachate on bacterial growth is also problematic.

Thank you for raising this point. We appreciate that drawing comparisons among studies with different concentrations is challenging. We have toned down the comparative language in the first paragraph of Discussion and explained that water characteristics may also be responsible for differences among studies. As for the specific comparison to Romera-Castillo et al. we are comparing the increase in BPP relative to added leachate concentration rather than BPP on its own.

Changed on lines 219-223:

“Therefore, we found strong effects of plastics at environmentally relevant concentrations, although differences between our study and others may be due to differences in the characteristics of background waters. These positive effects may disappear at higher leachate concentrations **and/or in different waters**, as found by Tetu et al.11 who added 1.3- to 250-times more plastic-derived DOM than us **into artificial seawater**.”

We have further addressed the point about using a single concentration in the new section on study limitations.

Added on lines 288-292

“Finally, our study used a single LDPE concentration that was representative of plastic concentrations found in lakes near population centres (Supplementary Methods 1). Higher

concentrations, such as found at waste management sites, may have less positive effects on microorganisms¹¹, especially if higher concentrations of toxic additives accumulate.”

Importantly, irrespective of the number of concentrations or sources we test, our study does more help to understand variation among past studies, because it shows that the water characteristics (and now) microbial composition matter in how bacteria use leachate.

The last concern I have is in relation to the non-specificity of the bacteria in your study. As you may be well aware, the growth of bacteria in response to different carbon sources could differ. Therefore, you may find that the leachate could potentially have different impacts on different bacterial communities. Therefore, making a conclusion on the impact of LDPE leachate on bacterial growth, in general, is also problematic without first ascertain if the impact is the same for different groups of bacteria.

We entirely agree that different bacteria may respond differently. We have made two major changes to address this comment. First, we now include bacterial diversity as a predictor in our statistical models for both BPP and BGE. Diversity had a statistically significant interaction with the plastic treatment for BGE and so we added a new paragraph to the Results section. Second, we identified individual bacteria taxa that were associated with the greatest increases in both BPP and BGE. This second change adds entirely new and important information to our paper – thank you for your encouraging us to do this with your comment.

Added on lines 402-429:

“In addition to DOM characteristics, we considered how the composition and diversity of bacteria influenced their responses to plastic leachate. To characterise bacterial communities, DNA was extracted from the Sterivex filters following an established protocol⁵⁴ with minor modifications. Briefly, we placed the filters, which were separated from the glass filtration unit under sterile conditions, into a cryotube containing silica and zirconia beads (3.0, 0.7, and 0.1 mm diameter) before vortexing at 2,850 rpm for 15 min. Then, we added 0.6 mL of phenol-chloroform-isopropanol (25:24:1), 0.6 mL of 5% cetrimonium bromide, 60 µl of 10% sodium dodecyl sulfate, and 60 µl 10% N-lauroylsarcosine and vortexed the solution at 2,850 rpm for 15 min. We then centrifuged the samples at 16 g for 15 min at 4°C and collected the supernatant. To the supernatant, we added an equal volume (ca. 0.6 mL) of chloroform-isopropanol (24:1), mixed the samples by inversion, and centrifuged at 16 g for 10 min at 4°C. We again collected the supernatant and precipitated the DNA at 4°C overnight in polyethylene glycol with 1.6M sodium chloride. We centrifuged the samples again at 17 g for 90 min at 4°C, removed the supernatant, and washed the pellet with ice-cold (-20°C) 70% ethanol. The DNA was dissolved in ultrapure water and quantified on a Qubit fluorometer (ThermoFisher, USA). We also extracted DNA from a ZymoBIOMICS™ Microbial Community Standard II (Log Distribution) (Zymo Research, USA) and nuclease-free water (Qiagen, Germany) to act as positive and negative controls, respectively. Libraries were prepared exactly like the lake samples.

We amplified the V6 and V8 regions of the 16S rRNA gene using the bacteria specific primers⁷¹ 5' ACGCGHNRAACCTTACC 3' and 5' ACGGGCRGTGWGTRCAA 3'. Samples were sequenced at 2×300 bp paired-end on an on an Illumina MiSeq at the Integrated Microbiome Resource (Halifax, Nova Scotia, Canada)⁷¹. No DNA was retrieved from the negative control and no contaminants were present in the positive control. We then removed the primers from the raw sequences using *cutadapt*⁷² and assigned taxonomy with the *DADA2* pipeline⁷³ and the Silva v132 database⁷⁴. Overall, 1.7 million reads were

classified into 2,148 amplicon sequence variants (ASVs), which represented 75% of the total raw reads, and we used these to compute the Shannon diversity index⁷⁵. The raw sequences have been deposited in the EBI database under accession number PRJEB49321.”

Added on lines 432-438:

“The effect of plastics on BPP and BGE were tested using linear mixed effects models. As both BPP and BGE were not normally distributed, they were natural log transformed before analysis. We then considered the following fixed predictors for each bacterial response: functional diversity of lake DOM, **bacterial diversity (Shannon index)**, DOC and **TN concentrations, lake water temperature, pH**, and latitude. The latter variable was included to control for differences in lake location, which is known to influence bacterial community composition⁷⁶, and so which we hypothesized may affect the overall bacterial response.”

Added on lines 451-458:

“We identified which ASVs were associated with changes in BPP and BGE after plastic leachate addition. We separately estimated the log₂-fold change in the relative abundance of each ASV relative to fold-increases in BGE or BPP by fitting separate negative binomial generalised linear models to read counts using the *DESeq* function in the *DESeq2*⁷⁹ R package. All ASVs with <100 reads were removed to avoid inferring correlations with rare taxa that may be subject to more stochastic variation in abundance. P-values were adjusted to correct for multiple comparisons with the Benjamini-Hochberg method⁷⁹ and considered statistically significant beneath a threshold of 0.05.”

Added on lines 172-191:

“Bacterial diversity affects the efficiency of plastic leachate usage

The effect of plastic leachate on BGE varied with the diversity of bacteria present in the lake, as expected if microbial community composition influenced the use of novel DOM sources. We retrieved 2,148 amplicon sequence variants (ASVs) across 20 lakes subjected to 16S amplicon sequencing. Community composition was dominated by the genera *Acinetobacter*, *Exiguobacterium*, and *Brevundimonas* (Fig. S4). We then summarised differences in bacterial diversity using the Shannon index, which ranged between 3.46 to 6.38 per lake, similar to other studies in northern waters. We found that bacterial diversity interacted with the plastic treatment to influence BGE (Fig. 3d). At high bacterial diversity, plastic leachate addition increased BGE 2.93 [1.71, 5.03] times from an estimated mean of 6.59 [3.84, 11.3] % to 19.3 [11.2, 33.1] %. There was no effect of plastic addition at low bacterial diversity with a 1.08 [0.58, 1.99] times difference. Bacterial diversity also had no effect on the response of BPP to leachate addition, as expected if taxa did not strongly discriminate in their use of labile, plastic-derived compounds, but instead used them with varying efficiency (Δ AIC from retaining interaction: 1.66).”

To identify which genera responded most strongly to the plastic leachate, we tested if some ASVs were more abundant when BPP and BGE increased after leachate addition. We found that the fold increases in BPP and BGE were positively correlated with 154 and 540 ASVs, respectively (Fig S4). ASVs that increased most in abundance with the fold increase in BPP and BGE belonged to the genera *Hymenobacter* and *Deinococcus*, respectively (Fig. S4).”

Added on lines 200-205:

“Figure 3. Bacterial growth efficiency (BGE) increased with the addition of plastic leachate depending on lake characteristics. (a) Bolded line shows mean \pm 95% CI for BGE in each treatment. Thin lines join mean effects for each of 18 study lakes with respiration data

(n = 1 replicate per treatment per lake). BGE increased relatively less with plastic leachate addition as either the (b) functional diversity of lake DOM increased, (c) lake DOC concentration increased, or (d) **lake bacterial diversity decreased.**”

Added on lines 57-68 of Supplementary Materials:

“**Figure S4. Use of plastic leachate positively correlates with abundances of different microbial taxa.** Each dot is the mean log₂-fold change in normalized abundance of a given genus with increasing fold change in (a) BGE or (b) BPP. Responses of individual ASVs were averaged at the genus-level, n = 286 and 78 ASVs for (a) and (b), respectively. (c) Mean relative abundance (percent of total reads) in each lake for the 20 most abundant genera. For all, error bars are the standard deviation when more than one ASV was included in a genus. Underlined genera were always associated with community-level metabolism, that is, occur in (a) and (b). Bold genera were the most abundant and so occurred in (c) and either (a) or (b). Colors denote phyla: Actinobacteriota (red), Bacteroidota (blue), Bdellovibrionota (orange), Cyanobacteria (green), Deinococcota (brown), Firmicutes (yellow), Proteobacteria (black), and Verrucomicrobiota (gold). ANPR = *Allorhizobium-Neorhizobium-Pararhizobium-Rhizobium.*”

Added on lines 249-257:

“Microbial diversity also influenced the increase in BGE after leachate addition. We specifically found that increases in BGE after leachate addition were amplified at higher levels of bacterial diversity. Greater diversity may increase the likelihood of taxa that can use plastic-derived compounds efficiently, thereby elevating the BGE of the entire community. To our knowledge, no study has explored how bacterial diversity influences the extent to which BGE responds to resource manipulation. Previous studies have instead correlated BGE to bacterial richness^{46,47}, and changes in BGE to bacterial community composition⁴⁸. More broadly, our results offer promise that some taxa may be particularly well suited to use plastic-derived compounds and remove them from the natural environment.”

Added on lines 263-279:

“Second, our results offer insight for efforts to identify environmental isolates that might remove plastic-derived compounds from the natural environment. Specifically, we found ASVs in the genera *Deinococcus* and *Hymenobacter* were associated with high levels of plastic leachate use, consistent with previous observations of microbial communities associated with biodegradable plastic films⁵¹. *Deinococcus* taxa have also been shown to match DNA sequences encoding a recently identified polyethylene terephthalatase enzyme from *Ideonella sakaiensis*⁵². Other taxa positively correlated with bacterial metabolism included *Exiguobacterium*, which were previously found to grown solely on polystyrene film (cite). However, bacteria capable of utilising leachates may differ from those that degrade plastic itself. Recent studies have isolated phylogenetically divergent bacteria with the ability to degrade plastics, including strains of Proteobacteria – such as *Pseudomonas* spp.⁵⁴⁻⁵⁶, Rhodobacteraceae⁵⁷, *Ideonella sakaiensis*⁵⁸, and *Acinetobacter baumannii*⁵⁹ – and Firmicutes such as *Bacillus* spp.^{54,56}. Many of these taxa were strongly associated with BPP and BGE in our study (Fig. S4). Irrespective of whether the microbes using leachates are the same as those decomposing it, the ability to uptake leachates is important for reducing chemical pollution from plastics^{11,38} and our results identifying taxa that do so can help direct biological remediation efforts.”

Based on these concerns, I suggest you add a study limitation section, where you discuss these limitations clearly.

We have added a study limitation section to the Discussion that discusses the first two general comments above. The Reviewer's third main comment on the non-specificity of bacteria in has been addressed through additional analyses as described above.

Added on lines 280-295:

“Our study has at least three limitations despite identifying clear effects of plastics on the metabolism of microbial communities. First, we focused solely on bacteria, but other microorganisms such as microalgae and fungi are also affected by plastics and plastic leachates⁶⁰⁻⁶³. These additional interactions may further influence the overall response of ecosystem metabolism to plastic pollution in addition to the effects of bacteria observed here. Second, we only leached LDPE. The chemical composition of leachate from other plastics will likely differ and so the type of plastic present within lakes may also influence bacteria alongside the local environment. However, LDPE is the most common plastic found in aquatic systems³⁵, so should contribute most to the DOM pool available for use by bacteria. Finally, our study used a single LDPE concentration that was representative of plastic concentrations found in lakes near population centres (Supplementary Methods 1). Higher concentrations, such as found at waste management sites, may have less positive effects on microorganisms¹¹, especially if higher concentrations of toxic additives accumulate. Irrespective, plastics will pollute the environment for decades⁶³. Our findings are therefore valuable as they suggest that some lakes (e.g. high DOC concentrations, functionally diverse DOM, low bacterial diversity) are least able to remove leachate dissolved from plastics and so would benefit most from future pollution management.”

Reviewer #3 (Remarks to the Author):

Potential negative impact of plastic pollution for ocean and freshwaters has been under discuss for last five to ten years. Meanwhile few studies (Yoshida et al. 2016, Taipale et al. 2019) has directly studied composition of (micro)plastics and found it to be relatively slow process, authors here study first time the impact of leachate from plastic bags. Unlike plastic decomposition you have found that leachates from plastic bags are readily usable and can support microbial activity. Moreover, you have found that leachates from plastic bags are more readily usable than DOM. Your incubation studies cover 29 lakes from Scandinavia and experiments are well planned and done. I keep that your finding will be important addition to plastic pollution literature.

We thank the Reviewer for this positive feedback.

My only suggestion for you is that you should more clearly separate plastic decomposition and utilization of leachates from plastic bags. Meanwhile leachates from plastic bags can be formed through abiotic process without microbial activity, plastic decomposition itself means microbial decomposition of plastic. These two processes are separate and meanwhile plastic itself is recalcitrant for microbial degradation, leachates from plastics can be labile and easily utilized by microbes. Therefore, microbes benefitting from leachates differs most likely those bacterial taxa decomposing plastic itself. Clear water lakes usually contain more labile organic carbon than humic waters, and thus microbes in humic waters can decompose complex organic compounds. You have reported here higher respond of microbes in low DOC than high DOC lake, which supports that leachates from plastic bags are easily degradable unlike plastic itself. This should be clearly mentioned in the abstract and debated in the discussion. Please, note also that microbes utilizing leachates are not necessarily same as plastic decomposers. Therefore I ask you kindly rephrase discussion that readers would not mix these two things.

We agree that our paper would benefit from distinguishing between bacteria that actually decompose plastics and bacteria that use the leachate. We have clarified this point in the Abstract and Discussion, largely informed by new analyses of microbial community composition data.

Added on lines 17-18:

“Abiotic and biotic degradation of plastics releases carbon-based substrates that are available for heterotrophic growth, but little is known about how these novel organic compounds influence microbial metabolism.”

Added on lines 271-279:

“However, bacteria capable of utilising leachates may differ from those that degrade plastic itself. Recent studies have isolated phylogenetically divergent bacteria with the ability to degrade plastics, including strains of Proteobacteria – such as *Pseudomonas* spp.⁵⁴⁻⁵⁶, Rhodobacteraceae⁵⁷, *Ideonella sakaiensis*⁵⁸, and *Acinetobacter baumannii*⁵⁹ – and Firmicutes such as *Bascillus* spp.^{54,56}. Many of these taxa were strongly associated with BPP and BGE in our study (Fig. S4). Irrespective of whether the microbes using leachates are the same as those decomposing it, the ability to uptake leachates is important for reducing chemical pollution from plastics^{11,38} and our results identifying taxa that do so can help direct biological remediation efforts.”

REVIEWERS' COMMENTS

Reviewer #1 (Remarks to the Author):

Thank you for the opportunity to revise again this work. I would like to thank the authors for providing such detailed and clear answers to the reviewers' questions and comments and for reconsidering some of their previous approaches and broadening their vision. The new data analysis and evaluation as well as the related discussion, have greatly improved the manuscript, also the final section commenting on the weakness of the study is very much appreciated. The article deserves to be published in its current format.

Reviewer #3 (Remarks to the Author):

I am delighted author's edition to the manuscript and clarifications on the text.